# Multiclass Total Variation Clustering

**Xavier Bresson**
University of Lausanne
Lausanne, Switzerland
xavier.bresson@unil.ch

**Thomas Laurent**
Loyola Marymount University
Los Angeles, CA 90045
tlaurent@lmu.edu

**David Uminsky**
University of San Francisco
San Francisco, CA 94117
duminsky@usfca.edu

**James H. von Brecht**
University of California, Los Angeles
Los Angeles, CA 90095
jub@math.ucla.edu

## Abstract

Ideas from the image processing literature have recently motivated a new set of clustering algorithms that rely on the concept of total variation. While these algorithms perform well for bi-partitioning tasks, their recursive extensions yield unimpressive results for multiclass clustering tasks. This paper presents a general framework for multiclass total variation clustering that does not rely on recursion. The results greatly outperform previous total variation algorithms and compare well with state-of-the-art NMF approaches.

## 1 Introduction

Many clustering models rely on the minimization of an energy over possible partitions of the data set. These discrete optimizations usually pose NP-hard problems, however. A natural resolution of this issue involves relaxing the discrete minimization space into a continuous one to obtain an easier minimization procedure. Many current algorithms, such as spectral clustering methods or non-negative matrix factorization (NMF) methods, follow this relaxation approach.

A fundamental problem arises when using this approach, however; in general the solution of the relaxed continuous problem and that of the discrete NP-hard problem can differ substantially. In other words, the relaxation is too loose. A *tight* relaxation, on the other hand, has a solution that closely matches the solution of the original discrete NP-hard problem. Ideas from the image processing literature have recently motivated a new set of algorithms [17, 18, 11, 12, 4, 15, 3, 2, 13, 10] that can obtain tighter relaxations than those used by NMF and spectral clustering. These new algorithms all rely on the concept of total variation. Total variation techniques promote the formation of sharp indicator functions in the continuous relaxation. These functions equal one on a subset of the graph, zero elsewhere and exhibit a non-smooth jump between these two regions. In contrast to the relaxations employed by spectral clustering and NMF, total variation techniques therefore lead to quasi-discrete solutions that closely resemble the discrete solution of the original NP-hard problem. They provide a promising set of clustering tools for precisely this reason.

Previous total variation algorithms obtain excellent results for two class partitioning problems [18, 11, 12, 3] . Until now, total variation techniques have relied upon a recursive bi-partitioning procedure to handle more than two classes. Unfortunately, these recursive extensions have yet to produce state-of-the-art results. This paper presents a general framework for multiclass total variation clustering that does not rely on a recursive procedure. Specifically, we introduce a new discrete multiclass clustering model, its corresponding continuous relaxation and a new algorithm for optimizing the relaxation. Our approach also easily adapts to handle either unsupervised or transductive

clustering tasks. The results significantly outperform previous total variation algorithms and compare well against state-of-the-art approaches [19, 20, 1]. We name our approach Multiclass Total Variation clustering (MTV clustering).

## 2  The Multiclass Balanced-Cut Model

Given a weighted graph $G = (V, W)$ we let $V = \{\mathbf{x}_1, \ldots, \mathbf{x}_N\}$ denote the vertex set and $W := \{w_{ij}\}_{1 \leq i,j \leq N}$ denote the non-negative, symmetric similarity matrix. Each entry $w_{ij}$ of $W$ encodes the similarity, or lack thereof, between a pair of vertices. The classical balanced-cut (or, Cheeger cut) [7, 8] asks for a partition of $V = A \cup A^c$ into two disjoint sets that minimizes the set energy

$$\mathrm{Bal}(A) := \frac{\mathrm{Cut}(A, A^c)}{\min\{|A|, |A^c|\}} = \frac{\sum_{\mathbf{x}_i \in A, \mathbf{x}_j \in A^c} w_{ij}}{\min\{|A|, |A^c|\}}. \tag{1}$$

A simple rationale motivates this model: clusters should exhibit similarity between data points, which is reflected by small values of $\mathrm{Cut}(A, A^c)$, and also form an approximately equal sized partition of the vertex set. Note that $\min\{|A|, |A^c|\}$ attains its maximum when $|A| = |A^c| = N/2$, so that for a given value of $\mathrm{Cut}(A, A^c)$ the minimum occurs when $A$ and $A^c$ have approximately equal size.

We generalize this model to the multiclass setting by pursuing the same rationale. For a given number of classes $R$ (that we assume to be known) we formulate our generalized balanced-cut problem as

$$\left. \begin{array}{c} \text{Minimize } \sum_{r=1}^{R} \dfrac{\mathrm{Cut}(A_r, A_r^c)}{\min\{\lambda|A_r|, |A_r^c|\}} \\[2ex] \text{over all disjoint partitions } A_r \cap A_s = \emptyset, \ A_1 \cup \cdots \cup A_R = V \text{ of the vertex set.} \end{array} \right\} \tag{P}$$

In this model the parameter $\lambda$ controls the sizes of the sets $A_r$ in the partition. Previous work [4] has used $\lambda = 1$ to obtain a multiclass energy by a straightforward sum of the two-class balanced-cut terms (1). While this follows the usual practice, it erroneously attempts to enforce that each set in the partition occupy half of the total number of vertices in the graph. We instead select the parameter $\lambda$ to ensure that each of the classes approximately occupy the appropriate fraction $1/R$ of the total number of vertices. As the maximum of $\min\{\lambda|A_r|, |A_r^c|\}$ occurs when $\lambda|A_r| = |A_r^c| = N - |A_r|$, we see that $\lambda = R - 1$ is the proper choice.

This general framework also easily incorporates *a priori* known information, such as a set of labels for transductive learning. If $L_r \subset V$ denotes a set of data points that are *a priori* known to belong to class $r$ then we simply enforce $L_r \subset A_r$ in the definition of an allowable partition of the vertex set. In other words, any allowable disjoint partition $A_r \cap A_s = \emptyset$, $A_1 \cup \cdots \cup A_R = V$ must also respect the given set of labels.

## 3  Total Variation and a Tight Continuous Relaxation

We derive our continuous optimization by relaxing the set energy (P) to the continuous energy

$$\mathcal{E}(F) = \sum_{r=1}^{R} \frac{\|f_r\|_{TV}}{\|f_r - \mathrm{med}_\lambda(f_r)\|_{1,\lambda}}. \tag{2}$$

Here $F := [f_1, \ldots, f_R] \in \mathbb{M}_{N \times R}([0, 1])$ denotes the $N \times R$ matrix that contains in its columns the relaxed optimization variables associated to the $R$ clusters. A few definitions will help clarify the meaning of this formula. The total variation $\|f\|_{TV}$ of a vertex function $f : V \to \mathbb{R}$ is defined by

$$\|f\|_{TV} = \sum_{i=1}^{n} \sum_{j=1}^{n} w_{ij} |f(\mathbf{x}_i) - f(\mathbf{x}_j)|. \tag{3}$$

Alternatively, if we view a vertex function $f$ as a vector $(f(\mathbf{x}_1), \ldots, f(\mathbf{x}_N))^t \in \mathbb{R}^N$ then we can write

$$\|f\|_{TV} := \|Kf\|_1. \tag{4}$$

Here $K \in \mathbb{M}_{M \times N}(\mathbb{R})$ denotes the *gradient matrix* of a graph with $M$ edges and $N$ vertices. Each row of $K$ corresponds to an edge and each column corresponds to a vertex. For any edge $(i, j)$ in the graph the corresponding row in the matrix $K$ has an entry $w_{ij}$ in the column corresponding to the $i^{\text{th}}$ vertex, an entry $-w_{ij}$ in the column corresponding to the $j^{\text{th}}$ vertex and zeros otherwise.

To make sense of the remainder of (2) we must introduce the asymmetric $\ell^1$-norm. This variant of the classical $\ell^1$-norm gives different weights to positive and negative values:

$$\|f\|_{1,\lambda} = \sum_{i=1}^{n} |f(\mathbf{x}_i)|_\lambda \qquad \text{where} \qquad |t|_\lambda = \begin{cases} \lambda t & \text{if } t \geq 0 \\ -t & \text{if } t < 0. \end{cases} \tag{5}$$

Finally we define the $\lambda$-median (or quantile), denoted $\text{med}_\lambda(f)$, as:

$$\text{med}_\lambda(f) = \text{the } (k+1)^{\text{st}} \text{ largest value in the range of } f, \text{ where } k = \lfloor N/(\lambda+1) \rfloor. \tag{6}$$

These definitions, as well as the relaxation (2) itself, were motivated by the following theorem. Its proof, in the supplementary material, relies only the three preceding definitions and some simple algebra.

**Theorem 1.** *If $f = \mathbf{1}_A$ is the indicator function of a subset $A \subset V$ then*

$$\frac{\|f\|_{TV}}{\|f - \text{med}_\lambda(f)\|_{1,\lambda}} = \frac{2 \, \text{Cut}(A, A^c)}{\min\{\lambda|A|, |A^c|\}}.$$

The preceding theorem allows us to restate the original set optimization problem (P) in the equivalent discrete form

$$\left. \begin{array}{c} \text{Minimize } \displaystyle\sum_{r=1}^{R} \frac{\|f_r\|_{TV}}{\|f_r - \text{med}_\lambda(f_r)\|_{1,\lambda}} \\[1em] \text{over non-zero functions } f_1, \ldots, f_R : V \to \{0, 1\} \text{ such that } f_1 + \ldots + f_R = \mathbf{1}_V. \end{array} \right\} \quad \text{(P')}$$

Indeed, since the non-zero functions $f_r$ can take only two values, zero or one, they must define indicator functions of some nonempty set. The simplex constraint $f_1 + \ldots + f_R = \mathbf{1}_V$ then guarantees that the sets $A_r := \{\mathbf{x}_i \in V : f_r(\mathbf{x}_i) = 1\}$ form a partition of the vertex set. We obtain the relaxed version (P-rlx) of (P') in the usual manner by allowing $f_r \in [0, 1]$ to have a continuous range. This yields

$$\left. \begin{array}{c} \text{Minimize } \displaystyle\sum_{r=1}^{R} \frac{\|f_r\|_{TV}}{\|f_r - \text{med}_\lambda(f_r)\|_{1,\lambda}} \\[1em] \text{over functions } f_1, \ldots, f_R : V \to [0, 1] \text{ such that } f_1 + \ldots + f_R = \mathbf{1}_V. \end{array} \right\} \quad \text{(P-rlx)}$$

The following two points form the foundation on which total variation clustering relies:

**1** — As the next subsection details, the total variation terms give rise to *quasi-indicator* functions. That is, the relaxed solutions $[f_1, \ldots, f_R]$ of (P-rlx) mostly take values near zero or one and exhibit a sharp, non-smooth transition between these two regions. Since these quasi-indicator functions essentially take values in the discrete set $\{0, 1\}$ rather than the continuous interval $[0, 1]$, solving (P-rlx) is almost equivalent to solving either (P) or (P'). In other words, (P-rlx) is a tight relaxation of (P).

**2** — Both functions $f \mapsto \|f\|_{TV}$ and $f \mapsto \|f - \text{med}_\lambda(f)\|_{1,\lambda}$ are convex. The simplex constraint in (P-rlx) is also convex. Therefore solving (P-rlx) amounts to minimizing a sum of ratios of convex functions with convex constraints. As the next section details, this fact allows us to use machinery from convex analysis to develop an efficient, novel algorithm for such problems.

### 3.1 The Role of Total Variation in the Formation of Quasi-Indicator Functions

To elucidate the precise role that the total variation plays in the formation of quasi-indicator functions, it proves useful to consider a version of (P-rlx) that uses a spectral relaxation in place of the total variation:

$$\left. \begin{array}{c} \text{Minimize } \displaystyle\sum_{r=1}^{R} \frac{\|f_r\|_{\text{Lap}}}{\|f_r - \text{med}_\lambda(f_r)\|_{1,\lambda}} \\[1em] \text{over functions } f_1, \ldots, f_R : V \to [0, 1] \text{ such that } f_1 + \ldots + f_R = \mathbf{1}_V \end{array} \right\} \quad \text{(P-rlx2)}$$

Here $\|f\|^2_{\text{Lap}} = \sum^n_{i=1} w_{ij}|f(\mathbf{x}_i) - f(\mathbf{x}_j)|^2$ denotes the spectral relaxation of $\text{Cut}(A, A^c)$; it equals $\langle f, Lf\rangle$ if $L$ denotes the unnormalized graph Laplacian matrix. Thus problem (P-rlx2) relates to spectral clustering (and therefore NMF [9]) with a positivity constraint. Note that the only difference between (P-rlx2) and (P-rlx) is that the exponent 2 appears in $\|\cdot\|_{\text{Lap}}$ while the exponent 1 appears in the total variation. This simple difference of exponent has an important consequence for the tightness of the relaxations. Figure 1 presents a simple example that illuminates this difference. If we bi-partition the depicted graph, i.e. a line with 20 vertices and edge weights $w_{i,i+1} = 1$, then the optimal cut lies between vertex 10 and vertex 11 since this gives a perfectly balanced cut. Figure 1(a) shows the vertex function $f_1$ generated by (P-rlx) while figure 1(b) shows the one generated by (P-rlx2). Observe that the solution of the total variation model coincides with the indicator function of the desired cut whereas the the spectral model prefers its smoothed version. Note that both functions in figure 1a) and 1b) have exactly the same total variation $\|f\|_{TV} = |f(\mathbf{x}_1) - f(\mathbf{x}_2)| + \cdots + |f(\mathbf{x}_{19}) - f(\mathbf{x}_{20})| = f(\mathbf{x}_1) - f(\mathbf{x}_{20}) = 1$ since both functions are monotonic. The total variation model will therefore prefer the sharp indicator function since it differs more from its $\lambda$-median than the smooth indicator function. Indeed, the denominator $\|f_r - \text{med}_\lambda(f_r)\|_{1,\lambda}$ is larger for the sharp indicator function than for the smooth one. A different scenario occurs when we replace the exponent one in $\|\cdot\|_{TV}$ by an exponent two, however. As $\|f\|^2_{\text{Lap}} = |f(\mathbf{x}_1) - f(\mathbf{x}_2)|^2 + \cdots + |f(\mathbf{x}_{19}) - f(\mathbf{x}_{20})|^2$ and $t^2 < t$ when $t < 1$ it follows that $\|f\|_{\text{Lap}}$ is much smaller for the smooth function than for the sharp one. Thus the spectral model will prefer the smooth indicator function despite the fact that it differs less from its $\lambda$-median. We therefore recognize the total variation as the driving force behind the formation of sharp indicator functions.

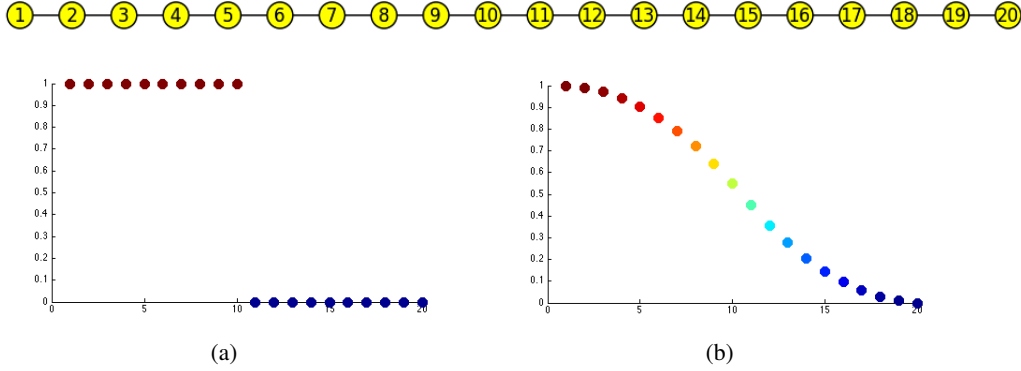

Figure 1: Top: The graph used for both relaxations. Bottom left: the solution given by the total variation relaxation. Bottom right: the solution given by the spectral relaxation. Position along the $x$-axis = vertex number, height along the $y$-axis = value of the vertex function.

This heuristic explanation on a simple, two-class example generalizes to the multiclass case and to real data sets (see figure 2). In simple terms, quasi-indicator functions arise due to the fact that the total variation of a sharp indicator function equals the total variation of a smoothed version of the same indicator function. The denominator $\|f_r - \text{med}_\lambda(f_r)\|_{1,\lambda}$ then measures the deviation of these functions from their $\lambda$-median. A sharp indicator function deviates more from its median than does its smoothed version since most of its values concentrate around zero and one. The energy is therefore much smaller for a sharp indicator function than for a smooth indicator function, and consequently the total variation clustering energy always prefers sharp indicator functions to smooth ones. For bi-partitioning problems this fact is well-known. Several previous works have proven that the relaxation is exact in the two-class case; that is, the total variation solution coincides with the solution of the original NP-hard problem [8, 18, 3, 5].

Figure 2 illustrates the result of the difference between total variation and NMF relaxations on the data set OPTDIGITS, which contains 5620 images of handwritten numerical digits. Figure 2(a) shows the quasi-indicator function $f_4$ obtained, before thresholding, by our MTV algorithm while 2(b) shows the function $f_4$ obtained from the NMF algorithm of [1]. We extract the portion of each function corresponding to the digits four and nine, then sort and plot the result. The MTV relaxation leads a sharp transition between the fours and the nines while the NMF relaxation leads to a smooth transition.

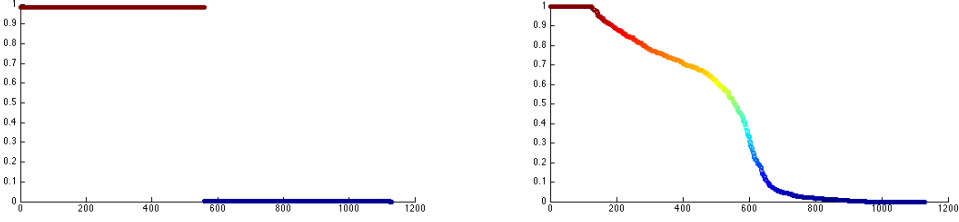

Figure 2: Left: Solution $f_4$ from our MTV algorithm (before thresholding) plotted over the fours and nines. Right: Solution $f_4$ from LSD [1] plotted over the fours and nines.

## 3.2 Transductive Framework

From a modeling point-of-view, the presence of transductive labels poses no additional difficulty. In addition to the simplex constraint

$$F \in \Sigma := \left\{ F \in \mathbb{M}_{N \times R}([0,1]) : f_r(\mathbf{x}_i) \geq 0, \ \sum_{r=1}^{R} f_r(\mathbf{x}_i) = 1 \right\} \tag{7}$$

required for unsupervised clustering we also impose the set of labels as a hard constraint. If $L_1, \ldots, L_R$ denote the $R$ vertex subsets representing the labeled points, so that $\mathbf{x}_i \in L_r$ means $\mathbf{x}_i$ belongs to class $r$, then we may enforce these labels by restricting $F$ to lie in the subset

$$F \in \Lambda := \left\{ F \in \mathbb{M}_{N \times R}([0,1]) : \forall r, \ (f_1(\mathbf{x}_i), \ldots, f_R(\mathbf{x}_i)) = \mathbf{e}_r \ \ \forall \ \mathbf{x}_i \in L_r \right\}. \tag{8}$$

Here $\mathbf{e}_r$ denotes the row vector containing a one in the $r^{\text{th}}$ location and zeros elsewhere. Our model for transductive classification then aims to solve the problem

$$\left. \text{Minimize} \ \sum_{r=1}^{R} \frac{\|f_r\|_{TV}}{\|f_r - \text{med}_\lambda(f_r)\|_{1,\lambda}} \ \text{ over matrices } F \in \Sigma \cap \Lambda. \right\} \tag{P-trans}$$

Note that $\Sigma \cap \Lambda$ also defines a convex set, so this minimization remains a sum of ratios of convex functions subject to a convex constraint. Transductive classification therefore poses no additional algorithmic difficulty, either. In particular, we may use the proximal splitting algorithm detailed in the next section for both unsupervised and transductive classification tasks.

# 4 Proximal Splitting Algorithm

This section details our proximal splitting algorithm for finding local minimizers of a sum of ratios of convex functions subject to a convex constraint. We start by showing in the first subsection that the functions

$$T(f) := \|f\|_{TV} \quad \text{and} \quad B(f) := \|f - \text{med}_\lambda(f)\mathbf{1}\|_{1,\lambda} \tag{9}$$

involved in (P-rlx) or (P-trans) are indeed convex. We also give an explicit formula for a subdifferential of $B$ since our proximal splitting algorithm requires this in explicit form. We then summarize a few properties of proximal operators before presenting the algorithm.

## 4.1 Convexity, Subgradients and Proximal Operators

Recall that we may view each function $f : V \to \mathbb{R}$ as a vector in $\mathbb{R}^N$ with $f(\mathbf{x}_i)$ as the $i^{\text{th}}$ component of the vector. We may then view $T$ and $B$ as functions from $\mathbb{R}^N$ to $\mathbb{R}$. The next theorem states that both $B$ and $T$ define convex functions on $\mathbb{R}^N$ and furnishes an element $v \in \partial B(f)$ by means of an easily computable formula. The formula for the subdifferential generalizes a related result for the symmetric case [11] to the asymmetric setting. We provide its proof in the supplementary material.

**Theorem 2.** *The functions $B$ and $T$ are convex. Moreover, given $f \in \mathbb{R}^N$ the vector $v \in \mathbb{R}^N$ defined below belongs to $\partial B(f)$:*

$$v(\mathbf{x}_i) = \begin{cases} \lambda & \text{if } f(\mathbf{x}_i) > \text{med}_\lambda(f) \\ \frac{n^- - \lambda n^+}{n^0} & \text{if } f(\mathbf{x}_i) = \text{med}_\lambda(f) \\ -1 & \text{if } f(\mathbf{x}_i) < \text{med}_\lambda(f) \end{cases} \quad \text{where} \quad \begin{cases} n^0 = |\{\mathbf{x}_i \in V : f(\mathbf{x}_i) = \text{med}_\lambda(f)\}| \\ n^- = |\{\mathbf{x}_i \in V : f(\mathbf{x}_i) < \text{med}_\lambda(f)\}| \\ n^+ = |\{\mathbf{x}_i \in V : f(\mathbf{x}_i) > \text{med}_\lambda(f)\}| \end{cases}$$

In the above theorem $\partial B(f)$ denotes the subdifferential of $B$ at $f$ and $v \in \partial B(f)$ denotes a subgradient. Given a convex function $A : \mathbb{R}^N \to \mathbb{R}$, the *proximal operator* of $A$ is defined by

$$\text{prox}_A(g) := \underset{f \in \mathbb{R}^N}{\text{argmin}} \ A(f) + \frac{1}{2} ||f - g||_2^2. \tag{10}$$

If we let $\delta_C$ denote the *barrier function* of the convex set $C$, that is

$$\delta_C(f) = 0 \text{ if } f \in C \text{ and } \delta_C(f) = +\infty \text{ if } f \notin C, \tag{11}$$

then we easily see that $\text{prox}_{\delta_C}$ is simply the least-squares projection on $C$, in other words, $\text{prox}_{\delta_C}(f) = \text{proj}_C(f) := \underset{g \in C}{\text{argmin}} \ \frac{1}{2} ||f - g||_2^2$. In this manner the proximal operator defines a mapping from $\mathbb{R}^N$ to $\mathbb{R}^N$ that generalizes the least-squares projection onto a convex set.

## 4.2 The Algorithm

We can rewrite the problem (P-rlx) or (P-trans) as

$$\text{Minimize} \quad \delta_C(F) + \sum_{r=1}^{R} E(f_r) \quad \text{over all matrices } F = [f_1, \ldots, f_r] \in \mathbb{M}_{N \times R} \tag{12}$$

where $E(f_r) = T(f_r)/B(f_r)$ denotes the energy of the quasi-indicator function of the $r^{\text{th}}$ cluster. The set $C = \Sigma$ or $C = \Sigma \cap \Lambda$ is the convex subset of $\mathbb{M}_{N \times R}$ that encodes the simplex constraint (7) or the simplex constraint with labels. The corresponding function $\delta_C(F)$, defined in (11), is the barrier function of the desired set. Beginning from an initial iterate $F^0 \in C$ we propose the following proximal splitting algorithm:

$$F^{k+1} := \text{prox}_{\mathcal{T}^k + \delta_C}(F^k + \partial \mathcal{B}^k(F^k)). \tag{13}$$

Here $\mathcal{T}^k(F)$ and $\mathcal{B}^k(F)$ denote the convex functions

$$\mathcal{T}^k(F) := \sum_{r=1}^{R} c_r^k \, T(f_r) \qquad \mathcal{B}^k(F) := \sum_{r=1}^{R} d_r^k \, B(f_r),$$

the constants $(c_r^k, d_r^k)$ are computed using the previous iterate

$$c_r^k = \Delta^k / B(f_r^k) \quad \text{and} \quad d_r^k = \Delta^k E(f_r^k)/B(f_r^k)$$

and $\Delta^k$ denotes the timestep for the current iteration. This choice of the constants $(c_r^k, d_r^k)$ yields $\mathcal{B}^k(F^k) = \mathcal{T}^k(F^k)$, and this fundamental property allows us to derive (see supplementary material) the energy descent estimate:

**Theorem 3** (Estimate of the energy descent). *Each of the $F^k$ belongs to $C$, and if $B_r^k \neq 0$ then*

$$\sum_{r=1}^{R} \frac{B_r^{k+1}}{B_r^k} \left( E_r^k - E_r^{k+1} \right) \geq \frac{||F^k - F^{k+1}||^2}{\Delta^k} \tag{14}$$

*where $B_r^k, E_r^k$ stand for $B(f_r^k), E(f_r^k)$.*

Inequality (14) states that the energies of the quasi-indicator functions (as a weighted sum) decrease at every step of the algorithm. It also gives a lower bound for how much these energies decrease. As the algorithm progress and the iterates stabilize the ratio $B_r^{k+1}/B_r^k$ converges to 1, in which case the sum, rather than a weighted sum, of the individual cluster energies decreases.

Our proximal splitting algorithm (13) requires two steps. The first step requires computing $G^k = F^k + \partial \mathcal{B}^k(F^k)$, and this is straightforward since theorem 2 provides the subdifferential of $B$, and therefore of $\mathcal{B}^k$, through an explicit formula. The second step requires computing $\text{prox}_{\mathcal{T}^k + \delta_C}(G^k)$, which seems daunting at a first glance. Fortunately, minimization problems of this form play an important role in the image processing literature. Recent years have therefore produced several fast and accurate algorithms for computing the proximal operator of the total variation. As $\mathcal{T}^k + \delta_C$ consists of a weighted sum of total variation terms subject to a convex constraint, we can readily adapt

these algorithms to compute the second step of our algorithm efficiently. In this work we use the primal-dual algorithm of [6] with acceleration. This relies on a proper uniformly convex formulation of the proximal minimization, which we detail completely in the supplementary material.

The primal-dual algorithm we use to compute $\text{prox}_{\mathcal{T}^k + \delta_C}(G^k)$ produces a sequence of approximate solutions by means of an iterative procedure. A stopping criterion is therefore needed to indicate when the current iterate approximates the actual solution $\text{prox}_{\mathcal{T}^k + \delta_C}(G^k)$ sufficiently. Ideally, we would like to terminate $F^{k+1} \approx \text{prox}_{\mathcal{T}^k + \delta_C}(G^k)$ in such a manner so that the energy descent property (14) still holds and $F^{k+1}$ always satisfies the required constraints. In theory we cannot guarantee that the energy estimate holds for an inexact solution. We may note, however, that a slightly weaker version of the energy estimate (14)

$$\sum_{r=1}^{R} \frac{B_r^{k+1}}{B_r^k} \left(E_r^k - E_r^{k+1}\right) \geq (1 - \epsilon) \frac{\|F^k - F^{k+1}\|^2}{\Delta^k} \tag{15}$$

holds after a finite number of iterations of the inner minimization. Moreover, this weaker version still guarantees that the energies of the quasi-indicator functions decrease as a weighted sum in exactly the same manner as before. In this way we can terminate the inner loop adaptively: we solve $F^{k+1} \approx \text{prox}_{\mathcal{T}^k + \delta_C}(G^k)$ less precisely when $F^{k+1}$ lies far from a minimum and more precisely as the sequence $\{F^k\}$ progresses. This leads to a substantial increase in efficiency of the full algorithm.

Our implementation of the proximal splitting algorithm also guarantees that $F^{k+1}$ always satisfies the required constraints. We accomplish this task by implementing the primal-dual algorithm in such a way that each inner iteration always satisfies the constraints. This requires computing the projection $\text{proj}_C(F)$ exactly at each inner iteration. The overall algorithm remains efficient provided we can compute this projection quickly. When $C = \Sigma$ the algorithm [14] performs the required projection in at most $R$ steps. When $C = \Sigma \cap \Lambda$ the computational effort actually decreases, since in this case the projection consists of a simplex projection on the unlabeled points and straightforward assignment on the labeled points. Overall, each iteration of the algorithm scales like $O(NR^2) + O(MR) + O(RN \log(N))$ for the simplex projection, application of the gradient matrix and the computation of the balance terms, respectively.

We may now summarize the full algorithm, including the proximal operator computation. In practice we find the choices $\Delta^k = \max\{B_1^k, \ldots, B_R^k\}$ and any small $\epsilon$ work well, so we present the algorithm with these choices. Recall the matrix $K$ in (4) denotes the gradient matrix of the graph.

---

**Algorithm 1** Proximal Splitting Algorithm

---

Input: $F \in C, P = 0, L = \|K\|_2, \tau = L^{-1}, \epsilon = 10^{-3}$
**while** loop not converged **do**
   *//Perform outer step $G^k = F^k + \partial \mathcal{B}^k(F^k)$*
   $\Delta = \max_r B(f_r)$  $\Delta_0 = \min_r B(f_r)$   $\sigma = \Delta_0^2 (\tau \Delta^2 L^2)^{-1}$   $\bar{F} = F$
   $D_E = \text{diag}\left[\frac{E(f_1)}{B(f_1)}, \ldots, \frac{E(f_R)}{B(f_R)}\right]$   $D_B = \text{diag}\left[\frac{\Delta}{B(f_1)}, \ldots, \frac{\Delta}{B(f_R)}\right]$
   $V = \Delta[\partial B(f_1), \ldots, \partial B(f_R)]D_E$ (using theorem 2)
   $G = F + V$
   *//Perform $F^{k+1} \approx prox_{\mathcal{T}^k + \delta_C}(G^k)$ until energy estimate holds*
   **while** (15) fails **do**
     $\tilde{P} = P + \sigma K \bar{F} D_B$   $P = \tilde{P}/\max\{|\tilde{P}|, 1\}$ (both operations entriwise)   $F_{\text{old}} = F$
     $\tilde{F} = F - \tau K^t P D_B$   $F = (\tilde{F} + \tau G)/(1 + \tau)$   $F = \text{proj}_C(F)$
     $\theta = 1/\sqrt{1 + 2\tau}$   $\tau = \theta\tau$   $\sigma = \sigma/\theta$   $\bar{F} = (1 + \theta)F - \theta F_{\text{old}}$
   **end while**
**end while**

---

# 5 Numerical Experiments

We now demonstrate the MTV algorithm for unsupervised and transductive clustering tasks. We selected six standard, large-scale data sets as a basis of comparison. We obtained the first data set

(4MOONS) and its similarity matrix from [4] and the remaining five data sets and matrices (WE-BKB4, OPTDIGITS, PENDIGITS, 20NEWS, MNIST) from [19]. The 4MOONS data set contains 4K points while the remaining five contain 4.2K, 5.6K, 11K, 20K and 70K points, respectively.

Our first set of experiments compares our MTV algorithm against other unsupervised approaches. We compare against two previous total variation algorithms [11, 3], which rely on recursive bi-partitioning, and two top NMF algorithms [1, 19]. We use the normalized Cheeger cut versions of [11] and [3] with default parameters. We used the code available from [19] to test each NMF algorithm. The non-recursive NMF algorithms (LSD [1], NMFR [19]) received two types of initial data: (a) the deterministic data used in [19]; (b) a random procedure leveraging normalized-cut [16]. Procedure (b) first selects one data point uniformly at random from each computed NCut cluster, then sets $f_r$ equal to one at the data point drawn from the $r^{\text{th}}$ cluster and zero otherwise. We then propagate this initial stage by replacing each $f_r$ with $(I+L)^{-1} f_r$ where $L$ denotes the unnormalized graph Laplacian. Finally, to aid the NMF algorithms, we add a small constant $0.2$ to the result (each performed better than without adding this constant). For MTV we use 30 random trials of (b) then report the cluster purity (c.f. [19] for a definition of purity) of the solution with the lowest discrete energy (P). We then use each NMF with exactly the same initial conditions and report simply the highest purity achieved over all 31 runs. This biases the results in favor of the NMF algorithms. Due to the non-convex nature of these algorithms, the random initialization gave the best results and significantly improved upon previously reported results of LSD in particular. For comparison with [19], initialization (a) is followed by 10,000 iterations of each NMF algorithm. Each trial of (b) is followed by 2000 iterations of each non-recursive algorithm. The following table reports the results. Our next set of experiments demonstrate our algorithm in a transductive setting. For each data set

| Alg/Data | 4MOONS | WEBKB4 | OPTDIGITS | PENDIGITS | 20NEWS | MNIST |
|---|---|---|---|---|---|---|
| NCC-TV [3] | 88.75 | 51.76 | 95.91 | 73.25 | 23.20 | 88.80 |
| 1SPEC [11] | 73.92 | 39.68 | 88.65 | 82.42 | 11.49 | 88.17 |
| LSD [1] | 99.40 | 54.50 | 97.94 | 88.44 | 41.25 | 95.67 |
| NMFR [19] | 77.80 | 64.32 | 97.92 | 91.21 | 63.93 | 96.99 |
| MTV | 99.53 | 59.15 | 98.29 | 89.06 | 39.40 | 97.60 |

we randomly sample either one label per class or a percentage of labels per class from the ground truth. We then run ten trials of initial condition (b) (propagating all labels instead of one) and report the purity of the lowest energy solution as before along with the average computational time (for simple MATLAB code running on a standard desktop) of the ten runs. We terminate the algorithm once the relative change in energy falls below $10^{-4}$ between outer steps of algorithm 1. The table below reports the results. Note that for well-constructed graphs (such as MNIST), our algorithm performs remarkably well with only one label per class.

| Labels | 4MOONS | WEBKB4 | OPTDIGITS | PENDIGITS | 20NEWS | MNIST |
|---|---|---|---|---|---|---|
| 1 | 99.55/ 3.0s | 56.58/ 1.8s | 98.29/ 7s | 89.17/ 14s | 50.07/ 52s | 97.53/ 98s |
| 1% | 99.55/ 3.1s | 58.75/ 2.0s | 98.29/ 4s | 93.73/ 9s | 61.70/ 54s | 97.59/ 54s |
| 2.5% | 99.55/ 1.9s | 57.01/ 1.7s | 98.35/ 3s | 95.83/ 7s | 67.61/ 42s | 97.72/ 39s |
| 5% | 99.53/ 1.2s | 58.34/ 1.3s | 98.38/ 2s | 97.98/ 5s | 70.51/ 32s | 97.79/ 31s |
| 10% | 99.55/ 0.8s | 62.01/ 1.2s | 98.45/ 2s | 98.22/ 4s | 73.97/ 25s | 98.05/ 25s |

Our non-recursive MTV algorithm vastly outperforms the two previous recursive total variation approaches and also compares well with state-of-the-art NMF approaches. Each of MTV, LSD and NMFR perform well on manifold data sets such as MNIST, but NMFR tends to perform best on noisy, non-manifold data sets. This results from the fact that NMFR uses a costly graph smoothing technique while our algorithm and LSD do not. We plan to incorporate such improvements into the total variation framework in future work. Lastly, we found procedure (b) can help overcome the lack of convexity inherent in many clustering approaches. We plan to pursue a more principled and efficient initialization along these lines in the future as well. Overall, our total variation framework presents a promising alternative to NMF methods due to its strong mathematical foundation and tight relaxation.

**Acknowledgements**: Supported by NSF grant DMS-1109805, AFOSR MURI grant FA9550-10-1-0569, ONR grant N000141210040, and Swiss National Science Foundation grant SNSF-141283.

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
