[Supplementary Material]

# Multiclass Total Variation Clustering
# (Supplementary Material)

**Xavier Bresson**
University of Lausanne
Lausanne, Switzerland
xavier.bresson@unil.ch

**Thomas Laurent**
Loyola Marymount University
Los Angeles, CA 90045
tlaurent@lmu.edu

**David Uminsky**
University of San Francisco
San Francisco, CA 94117
duminsky@usfca.edu

**James H. von Brecht**
University of California, Los Angeles
Los Angeles, CA 90095
jub@math.ucla.edu

## 1 Proofs of Theorems

**Theorem 1.** *If $f = \mathbf{1}_A$ is the indicator function of a subset $A \subset V$ then*

$$\frac{\|f\|_{TV}}{\|f - \mathrm{med}_\lambda(f)\|_{1,\lambda}} = \frac{2\,\mathrm{Cut}(A, A^c)}{\min\{\lambda|A|, |A^c|\}}.$$

*Proof.* The fact that $\|f\|_{TV} = 2\,\mathrm{Cut}(A, A^c)$ follows directly from the definition of the total variation. Indeed, a straightforward computation shows

$$\|f\|_{TV} = \sum_{\mathbf{x}_i \in A} \sum_{j=1}^{N} w_{ij}|1 - f(\mathbf{x}_j)| + \sum_{\mathbf{x}_i \in A^c} \sum_{j=1}^{N} w_{ij}|f(\mathbf{x}_j)| = \sum_{\mathbf{x}_i \in A} \sum_{\mathbf{x}_j \in A^c} w_{ij} + \sum_{\mathbf{x}_i \in A^c} \sum_{\mathbf{x}_j \in A} w_{ij}.$$

Thus $\|f\|_{TV} = 2\,\mathrm{Cut}(A, A^c)$ as $W$ is symmetric. Let $B(f) := \|f - \mathrm{med}_\lambda(f)\|_{1,\lambda}$. To show that $B(f) = \min\{\lambda|A|, |A^c|\}$, suppose first that $\lambda|A| \leq |A^c|$. This inequality implies $\lambda|A| \leq N - |A|$, or equivalently that $|A| \leq N/(1 + \lambda)$. Thus $|A| \leq k := \lfloor N/(1 + \lambda) \rfloor$, and since $f = \mathbf{1}_A$ for $|A| \leq k$ it follows immediately that the $(k+1)^{\text{st}}$ largest entry in the range of $f$ equals zero. Thus $\mathrm{med}_\lambda(f) = 0$ by definition. A direct computation then yields that $B(f) = \sum_{i \in V} |f(\mathbf{x}_i)|_\lambda = \lambda|A|$. In the converse case, the fact that $|A^c| < \lambda|A|$ implies $|A| > N/(1 + \lambda) \geq k$. Thus $|A| \geq k + 1$ and $\mathrm{med}_\lambda(f) = 1$. Direct computation then shows that $B(f) = \sum_{i \in V} |f(\mathbf{x}_i) - 1|_\lambda = |A^c|$ as claimed. □

**Lemma 1.** *Let $h \in \mathbb{R}^N$ and suppose $v \in \mathbb{R}^N$ satisfies*

$$v(\mathbf{x}_i) \in \begin{cases} \lambda & \text{if } h(\mathbf{x}_i) > 0 \\ [-1, \lambda] & \text{if } h(\mathbf{x}_i) = 0 \\ -1 & \text{if } h(\mathbf{x}_i) < 0. \end{cases} \tag{1}$$

*Then $v \in \partial \|h\|_{1,\lambda}$.*

*Proof.* Note that $|h(\mathbf{x}_i)|_\lambda = v(\mathbf{x}_i)h(\mathbf{x}_i)$ for each $\mathbf{x}_i$, so that for arbitrary $g \in \mathbb{R}^N$ and each $\mathbf{x}_i$ the inequality

$$|g(\mathbf{x}_i)|_\lambda - |h(\mathbf{x}_i)|_\lambda \geq v(\mathbf{x}_i)\,(g(\mathbf{x}_i) - h(\mathbf{x}_i))$$

holds. Summing both sides over all $\mathbf{x}_i \in V$ then gives the claim. □

**Theorem 2.** *The functions $B$ and $T$ are convex. Moreover, given $f \in \mathbb{R}^N$ the vector $v \in \mathbb{R}^N$ defined by*

$$v(\mathbf{x}_i) = \begin{cases} \lambda & \text{if } f(\mathbf{x}_i) > \text{med}_\lambda(f) \\ \frac{n^- - \lambda n^+}{n^0} & \text{if } f(\mathbf{x}_i) = \text{med}_\lambda(f) \\ -1 & \text{if } f(\mathbf{x}_i) < \text{med}_\lambda(f) \end{cases} \quad \text{where} \quad \begin{cases} n^0 = |\{\mathbf{x}_i \in V : f(\mathbf{x}_i) = \text{med}_\lambda(f)\}| \\ n^- = |\{\mathbf{x}_i \in V : f(\mathbf{x}_i) < \text{med}_\lambda(f)\}| \\ n^+ = |\{\mathbf{x}_i \in V : f(\mathbf{x}_i) > \text{med}_\lambda(f)\}| \end{cases}$$

*belongs to $\partial B(f)$.*

*Proof.* The convexity of $T(f)$ follows directly from its definition and a straightforward computation using the definition of convexity. Due to the continuity $B(f)$, to show convexity it suffices to establish the existence of a subdifferential at every point.

To this end note that $\text{med}_\lambda(f) \in \text{range}(f)$, so that in particular $n^0 \geq 1$ by definition. Let $1 \leq k := \lfloor N/(1 + \lambda) \rfloor < N$ denote that entry of $f$ so that $f(\mathbf{x}_k) = \text{med}_\lambda(f)$. By definition of $\text{med}_\lambda(f)$ there exist at most $k$ elements of $f$ larger than $\text{med}_\lambda(f)$, so that $n^+ \leq k \leq N/(1 + \lambda)$. As $N = n^- + n^0 + n^+$ this implies $\frac{\lambda n^+ - n^-}{n^0} \leq 1$. Similarly there exist at most $N - (k + 1)$ elements of $f$ smaller than $\text{med}_\lambda(f)$, so that $n^- \leq N - (k + 1) \leq N - N/(1 + \lambda)$. The fact that $N = n^- + n^0 + n^+$ then implies $\frac{n^- - \lambda n^+}{n^0} \leq \lambda$. Combining this with the previous inequality yields $-1 \leq \frac{n^- - \lambda n^+}{n^0} \leq \lambda$.

Put $h := f - \text{med}_\lambda(f)\mathbf{1}$, and note that the vector $v$ defined above satisfies $v \in \partial \|h\|_{1,\lambda}$ by the preceeding lemma. Thus for any $g \in \mathbb{R}^N$ it holds that

$$\|g - \text{med}_\lambda(g)\mathbf{1}\|_{1,\lambda} - \|f - \text{med}_\lambda(f)\mathbf{1}\|_{1,\lambda} \geq \langle v, g - f + (\text{med}_\lambda(f) - \text{med}_\lambda(g))\mathbf{1} \rangle$$

by definition of the subdifferential. Note also that $\langle v, \mathbf{1} \rangle = 0$, so that in fact

$$B(g) - B(f) = \|g - \text{med}_\lambda(g)\mathbf{1}\|_{1,\lambda} - \|f - \text{med}_\lambda(f)\mathbf{1}\|_{1,\lambda} \geq \langle v, g - f \rangle$$

for $g \in \mathbb{R}^N$ arbitrary. Thus $v \in \partial B(f)$ by definition of the subdifferential. In particular $\partial B(f)$ is always non-empty, so $B(f)$ is convex. $\qquad\square$

**Theorem 3** (Estimate of the energy descent). *Each of the $F^k$ belongs to $C$, and if $B_r^k \neq 0$ then*

$$\sum_{r=1}^R \frac{B_r^{k+1}}{B_r^k} \left( E_r^k - E_r^{k+1} \right) \geq \frac{\|F^k - F^{k+1}\|^2}{\Delta^k} \tag{2}$$

*where $B_r^k, E_r^k$ stand for $B(f_r^k), E(f_r^k)$.*

*Proof.* Let $V^k \in \partial \mathcal{B}^k(F^k)$. Then by definition of the subdifferential it follows that

$$\mathcal{B}^k(F^{k+1}) \geq \mathcal{B}^k(F^k) + \langle F^{k+1} - F^k, V^k \rangle. \tag{3}$$

As $F^{k+1} = \text{prox}_{\mathcal{T}^k + \delta_C}(F^k + V^k)$ the definition of the proximal operator implies that $F^{k+1} \in C$ and that also

$$F^k + V^k - F^{k+1} \in \partial(\mathcal{T}^k + \delta_C)(F^{k+1}).$$

The definition of the subdifferential and the fact that $\delta_C(F^k) = \delta_C(F^{k+1}) = 0$ then combine to imply

$$\mathcal{T}^k(F^k) \geq \mathcal{T}^k(F^{k+1}) + \langle F^k - F^{k+1}, F^k + V^k - F^{k+1} \rangle$$
$$= \mathcal{T}^k(F^{k+1}) + \|F^k - F^{k+1}\|^2 + \langle F^k - F^{k+1}, V^k \rangle \tag{4}$$

Adding (3) and (4) yields

$$\mathcal{T}^k(F^k) + \mathcal{B}^k(F^{k+1}) \geq \mathcal{T}^k(F^{k+1}) + \mathcal{B}^k(F^k) + \|F^k - F^{k+1}\|^2,$$

or equivalently that $\mathcal{B}^k(F^{k+1}) \geq \mathcal{T}^k(F^{k+1}) + \|F^k - F^{k+1}\|^2$ since $\mathcal{B}^k(F^k) = \mathcal{T}^k(F^k)$ by construction. Expanding this last inequality shows

$$\sum_{r=1}^R \frac{\Delta^k}{B_r^k} \left( E_r^k B_r^{k+1} - T_r^{k+1} \right) \geq \|F^k - F^{k+1}\|^2,$$

which yields the claim after by $B_r^{k+1}$ in each term of the summation. $\qquad\square$

## 2   Primal-Dual Formulation

Consider the minimization

$$F^{k+1} := \text{prox}_{\mathcal{T}^k + \delta_C}(G^k).$$

We may write this as the saddle-point problem

$$\min_{u \in \mathbb{R}^{NR}} \max_{p \in \mathbb{R}^{MR}} \langle p, \mathcal{K}u \rangle + G(u) - F^*(p).$$

Here the vector $u = (f_1, \ldots, f_R)^t$ is a "vectorized" version of $F$ and the matrix $\mathcal{K}$ denotes the block diagonal matrix

$$\mathcal{K} := \text{blkdiag}\left( \frac{\Delta^k}{B_1^k}K, \ldots, \frac{\Delta^k}{B_R^k}K \right)$$

where $K$ is the gradient matrix of the graph. We define the convex function $G(u)$ as

$$G(u) := \frac{1}{2} \sum_{r=1}^{R} ||f_r - g_r^k||^2 + \delta_C(u),$$

where $\delta_C$ denotes the barrier function of the convex set $C$ (either the simplex or simplex with labels) as before. The convex function $F^*(p)$ denotes the barrier function of the $l^\infty$ unit ball, so that

$$F^*(p) = \begin{cases} 0 & \text{if} \quad |p_i| \leq 1 \ \forall\, 1 \leq i \leq MR \\ +\infty & \text{otherwise.} \end{cases}$$

Note also that $G(u)$ is uniformly convex, in that if $v \in \partial G(u)$ denotes any subdifferential then for any $u' \in \mathbb{R}^{NR}$ the inequality

$$G(u') - G(u) \geq \langle v, u' - u \rangle + \frac{1}{2}||u - u'||^2$$

holds. We may therefore apply algorithm 2 of [1] with $\gamma = 1$ with to solve the saddle-point problem. This algorithm consists in the iterations

$$p^{n+1} = \text{prox}_{\sigma^n F^*}(p^n + \sigma^n \mathcal{K}\bar{u}^n)$$
$$u^{n+1} = \text{prox}_{\tau^n G}(u^n - \tau^n \mathcal{K}^t p^{n+1})$$
$$\theta^n = \frac{1}{\sqrt{1 + 2\tau^n}} \quad \tau^{n+1} = \theta^n \tau^n \quad \sigma^{n+1} = \sigma^n/\theta^n$$
$$\bar{u}^{n+1} = u^{n+1} + \theta^n(u^{n+1} - u^n)$$

and converges provided the inequality $\sigma^0 \leq (\tau^0 ||\mathcal{K}||_2^2)^{-1}$ holds for the initial timesteps. We may compute the inner proximal operators analytically to find

$$(\text{prox}_{\sigma^n F^*}(z))_i = z_i / \max\{1, |z_i|\} \quad \forall\, 1 \leq i \leq MR,$$

and by completing the square that

$$\text{prox}_{\tau^n G}(z) = \text{proj}_C\left( \frac{z + \tau^n g}{1 + \tau^n} \right),$$

where $g = (g_1^k, \ldots, g_R^k)^t$ denotes $G^k$ in vectorized form. The inner loop of algorithm 1 then follows by re-writing these computations in matrix form.

## References

[1] A. Chambolle and T. Pock. A First-Order Primal-Dual Algorithm for Convex Problems with Applications to Imaging. *Journal of Mathematical Imaging and Vision*, 40(1):120–145, 2011.