[Reviews · NeurIPS 2013]

Submitted by Assigned_Reviewer_4

The authors study a variant of ratio cut with R
clusters where the balancing function is biased towards
partitions where each cluster has the same size. The main
contribution of the paper is a continuous formulation and
an algorithm to optimize the criterion directly, whereas
previous algorithms are mostly limited to recursive splitting.

The direct solution of multi-cut problems instead of using
recursive splitting is an important problem given the new
developments in finding balanced graph cuts [3,4,5,11,12,18].

The authors first describe the discrete problem (P) and then
derive a relaxation of the problem (P-rlx). Opposite to
previous work [11,12,17,18] for the two cluster case, they cannot show that
this relaxation is exact, that is the discrete and continuous problems are
equivalent. While they argue that the relaxation is very good, this
is unfortunately not necessarily true. If there exists a minimizer
f=ones_A of the *single* ratio problem which is far better than all other
ones (in the extreme case A and A^c are disconnected), then the
solution of (P-rlx) is just f_r=c_r 1_A or f_r=c_r 1_A^c and
the c_r are chosen such that they sum up to one for every vertex
(note that rescaling of f_r does not change the objective).
Thus the condition that F generates a partition is *not* enforced by
the simplex constraint. In the extreme case where there exists a set
A which is disconnected, then the optimal value of the relaxation (P-rlx)
is zero whereas the optimal value of the discrete problem (P) clearly
need not be zero - thus the relaxation can be arbitrarily bad.
A more careful discussion is needed
rather than just arguing that the total variation enforces indicator
functions in Section 3.1. - this is well-known already in image processing
and can be made precise using the relation between submodular set functions
and their Lovasz extensions and was the basis of the previous work [3,4,5,11,12,18]
to prove that the relaxation is exact. The new claim that the simplex constraint
enforces that the sets form a partition is not shown and it is impossible to show as the above counterexample suggests.

The algorithm in 4.2. looks similar to previous work and the integration of the simplex constraint is rather standard. I cannot see how the estimate in Theorem 3 is related to descent of the total energy - the authors argue that the ratio
B_r^{k+1}/B_r^k tends to one and thus the total energy decreases. However, this
is not proven anywhere. It is not even clear if the algorithm does converge.

The experimental results are good even though [19] seems to perform better.
However, the actual point of the paper that MTV leads to better normalized cuts
is not documented, as just clustering errors are reported - at least for the previous
methods using recursive splitting [3,11] and MTV one could have shown also the
normalized cut values.

Pro:
- direct tackling multi-normalized-cut is an important problem
- new algorithm
- good experimental results

Con:
- relaxation is not tight as argued by the authors - it can be arbitrarily bad
- no convergence proof for the algorithm or even descent is guaranteed

------------------------------------------------------------------------------
Added after author response:

Tightness:
The suggested relaxation is not tight as the example in the review shows - and
this problem will remain even if the graph is connected and there is one dominating
cut. In the case where the graph is disconnected the suggested relaxation has value zero
whereas the original discrete problem need not. Also the spectral relaxation would yield
a better relaxation as it sums up the first k eigenvalues (where k is the number of clusters)
which would be non-zero (say k=10 and the graph has two connected components). This argument
holds even for criterion (P) as one can easily lower- and upper bound it in terms of (1).

The argument that the algorithm anyway just finds a solution close to an initial starting
point is very weak. If this is the case then the suggested algorithm is very bad. Moreover,
then the problem is transferred to finding a good initial solution which is as difficult
as the original problem and also not discussed in the paper.


Monotonicity and Convergence:
If these results are available then it is unclear why they were not put into the paper/supplementary material.
Anyway even if the algorithm has better properties as reported in the paper, the problems
with the continuous relaxation itself remain.

Purity:
To report purity is definitely fine. However, a comparison to recursive splitting techniques [3,11]
or standard spectral clustering (with k-means in the embedding space) could have been done both with
respect to criterion (1) and the suggested criterion (P), in order to judge if the suggested algorithm
really produces better cuts - as this is the only criterion one has in practice to judge a given
partition. It would even be interesting to see the performance of the NMF methods there.

Summary: The paper tackles an important problem and shows good experimental results, but the suggested relaxation can be arbitrarily bad and the convergence/descent properties of the algorithm are unclear.

Submitted by Assigned_Reviewer_5

Summary: The authors present a multiclass clustering algorithm based on total variation. The paper elegantly relates a tight relaxation of the balanced cut minimization task to the minimization of a sum of ratios of convex functions with convex constraints. The presented algorithm is a significant advance over existing total variation algorithms and this is demonstrated by an analysis using standard data sets of reasonably large scale.

Quality: The paper is of high technical quality; the proposed algorithm has solid theoretical underpinnings and is efficient and competitive with state-of-the-art approaches.

The numerical results provide a meaningful comparison to existing methods, demonstrating that the proposed method outperforms other total variation methods but is merely competitive with other state-of-the-art techniques. I think it would be useful to define cluster purity in Section 5 or at least provide a reference to a relevant paper where the metric is clearly defined ([19] for example). A comparison based on the metric that the algorithms are attempting to optimize would be useful.

The paper provides no discussion regarding how to choose the number of clusters. Although it is reasonable to consider that to be a separate issue, the paper should make it clearer that it assumes that an appropriate R is pre-specified.

Some (very brief) further discussion on scalability would be welcome. Many data sets of interest have hundreds of thousands or even millions of points and the paper does not make clear how the computational burden scales and what are the key contributing factors.

Clarity: The paper is well-written. The development of the method is logical; the proof of the key theorems are succinct and clear.

Originality: The proposed tight relaxation of the balanced cut optimization problem and its formulation as a sum of ratios of convex functions with convex constraints represent useful original research contributions.

Significance: The paper represents an improvement in the field of clustering algorithms based on total variation and it has the potential to have an impact in the field of research dedicated to this topic.
Summary: The paper makes a useful original research contribution in the field of clustering algorithms based on total variation, leading to an efficient multi-class clustering procedure that outperforms existing total variation approaches. More detail about the computation complexity and convergence properties of the algorithm would enhance the contribution.


Submitted by Assigned_Reviewer_7

This paper proposes a new multi-class formulation of total-variation based clustering, which solves some of the problems of previous methods based on the recursive application of binary clustering methods. The paper is solid and extends the current state-of-the-art in its topic, although it can hardly be considered a breakthrough. The text and mathematical formulations are well presented and clear. The experimental evaluation shows that the proposed method outperforms the previous recursive technique; however, with respect to other types of methods, the new scheme is competitive, but no more than that.
Summary: A good paper, with a solid contribution to the toolbox of data clustering algorithms. Some, but not much, significance and impact, since this is essentially an extension of previous work.
Author Feedback

Author rebuttal: We thank the reviewers for their helpful comments. The present work aims to communicate the general ideas behind our algorithm, and specifically our motivation for using total variation over NMF, to a wide audience that may not know of total variation techniques. This fact, as well as space limitations, led us to make a few editorial decisions regarding more technical aspects of the work. We feel a fuller discussion of these technical points belongs in a journal paper aimed at specialists, and that such a mathematical investigation is worthwhile only after a demonstration of state-of-the-art results. We now clarify a few technical points in light of the reviews.

Tightness: We discuss tightness as it relates to the advantages of using total variation relaxations over NMF and spectral ones. Specifically, our algorithm provides a tight relaxation in the sense that total variation promotes the formation of indicator functions while the other relaxations do not. The TV relaxation therefore more closely matches the solution space of the discrete problem. We show experimentally that TV does give a tighter relaxation than NMF (fig. 1 and 2) in this sense. We aim to give an intuitive, and not overly technical, explanation for why we can expect this to happen in the multiclass case even though the two-class arguments [18,11,12,3] break down. The graphs that pose theoretical difficulties for the simplex constraint, as described in the review, pose no difficulty in practice; a reasonable choice of initial data leads to a meaningful clustering rather than the pathological one described in the example. This difficulty also exists in theory for recent, state-of-the-art NMF methods (e.g. [1]) that nevertheless work well in practice if given a good initialization.

Computational complexity: The algorithm scales roughly like other iterative multiclass algorithms that use sparse similarity matrices do. The main contributing factors in the computational burden of each step are the projection to the simplex, the application of the gradient matrix K and the computation of the balance term. The simplex projection scales like O(N*R*R) in the worst case, and it parallelizes easily as the projection along each vertex decouples. Applying K scales like O(M*R) where M denotes the number of edges in the graph. The computation of the balance term requires sorting, so this step scales like O(R*N*log(N)) in the worst case. We agree a brief discussion of these issues would enhance the paper.

Monotonicity and convergence: Our proximal splitting algorithm has the same properties as the two-class total variation algorithms. A complication arises in the multiclass setting that is absent in the two-class case, namely that these results require a restriction on the time-step. By using an adaptive time-step in the algorithm, which can be computed in terms of Lipschitz constants of the total variation and balance terms, the estimate (15) implies that the algorithm is monotonic. This estimate combines with compactness of the simplex and closedness of the set-valued mapping defined by the algorithm to give convergence results. Specifically, all limit points of the iterative sequence are critical points of the continuous energy and local minima exhibit Lyapunov stability---the sequence will locally converge to isolated local minima given a sufficiently close initial guess. The proofs adapt ideas from [3], but these arguments, even just for monotonicity, prove exceedingly complicated and technical. In our implementation we suggest to use a simpler choice of time-step since it leads to slightly faster computations than using the provably monotonic time-step. Monotonicity still holds in practice with the simpler time-step, and we presented this choice as we felt it led to a more useful algorithm. We feel the convergence results and a comparison between different parameter choices in the algorithm belong in a longer, journal paper and inclusion of these details would obfuscate the message of the paper.

Metric for comparison between methods: We follow [19] and use purity due to the fact that each algorithm attempts to minimize a different energy. Our algorithm attempts to minimize a new discrete energy that has not appeared in the literature, so comparing based upon this criterion seemed unfair to the other methods. In particular, unlike the recursive total variation algorithms our algorithm does NOT aim to obtain lower normalized cut values as the review claims. The discrete energy we introduce correlates better with purity, at least on the examples we included, than the classical normalized cut energy does. The discrete model itself therefore has as much influence over the results as the continuous relaxation and algorithm do.

A few minor points: The review expresses the concern that the "new claim that the simplex constraint enforces that the sets form a partition is not shown and it is impossible to show." We never made such a claim. We state that the simplex constraint enforces that the sets form a partition at the DISCRETE level (the sentence to this effect appears BEFORE we introduce the continuous relaxation), i.e. referring to indicator functions of non-empty vertex subsets. Our statement is both true and obvious since we refer only to the discrete setting. The review correctly states that we neglected to mention that we consider the number of clusters as given. We will clarify this in the final version and also provide a reference for the definition of purity.

Significance: The present work represents the first demonstration that total variation clustering techniques compare well with the state-of-the-art.